# Social and Family Factors as Determinants of Sleep Habits in Japanese Elementary School Children: A Cross-Sectional Study from the Super Shokuiku School Project

**DOI:** 10.3390/children8020110

**Published:** 2021-02-05

**Authors:** Satomi Sawa, Michikazu Sekine, Masaaki Yamada

**Affiliations:** 1School of Human Development, University of Toyama, 3190 Gofuku, Toyama 930-8555, Japan; 2Department of Epidemiology and Health Policy, School of Medicine, University of Toyama, 2630 Sugitani, Toyama 930-0194, Japan; sekine@med.u-toyama.ac.jp (M.S.); masaakit@med.u-toyama.ac.jp (M.Y.)

**Keywords:** sleep habits, parental lifestyle, social background, breakfast consumption, screen time, physical activity

## Abstract

This study explored the associations of lifestyle, familial, and social factors with sleep habits in 1882 elementary school children, aged 6–13 years, from the Super Shokuiku School Project in January 2016. A survey assessed sex, grade, sleep habits, lifestyle, social background, and parental lifestyle. Bedtime “≥22:00,” wake-up time “≥07:00,” sleep duration “<8 h,” and “daytime sleepiness” were defined as poor sleep habits; correlates were analyzed using logistic regression. Skipping breakfast was consistently significantly associated with poor sleep, especially among children with late wake-up times (adjusted odds ratio 5.45; 95% confidence interval 3.20–9.30). Excessive screen time was associated with late bed and wake-up times. Physical inactivity was significantly associated with daytime sleepiness. Children of mothers with poor lifestyle habits were likely to go to bed late and feel sleepy the next day. Social and family factors were associated with children’s sleep habits. Several behaviors, including skipping breakfast, excessive screen time, and physical inactivity, were associated with poor sleep habits, manifesting as a night-oriented lifestyle. Although a longitudinal study is needed to determine causality, in addition to sleep education for children, sleep education for parents and society at large may be necessary to improve children’s sleep habits.

## 1. Introduction

Children are going to sleep late and sleeping less, paralleling the general population’s increasingly nocturnal lifestyles [1,2]. According to the 2017 National Health and Nutrition Survey, about 40% of Japanese adults average less than six hours of sleep daily [3]. Children also ranked last among the 17 countries surveyed regarding infants’ and toddlers’ sleep duration [4]. Furthermore, childhood and adolescent late bedtime and the decrease in sleep time are remarkable, resulting in an increase in daytime sleepiness and sleep deprivation [5].

Children’s sleep habits are established early and can later predispose them to lifestyle-related diseases [6,7]. Among children who went to bed after 22:00 at age three, 42% still did so in 4th grade (age 9–10), and they were 1.6-fold more likely to be obese in 7th grade (age 12–13) [6]. Poor sleep habits contribute to impairing normal brain development [8], school absence [9], and mental or psychosocial problems in elementary school children [10]. There is also the potential negative impact of television, video games, and the internet before bedtime and the possibility that late after-school activities can disturb sleep/wake patterns [11]. In short, elementary school is the best time to examine sleep habits in relation to other lifestyle factors and living environment factors and intervene to correct their life rhythm.

Children’s lifestyle habits are related to familial and social factors, such as parental habits and affluence [12,13,14,15,16,17,18,19]. Children’s sleep habits are also influenced by their families; children’s night-oriented lifestyle has been attributed to family members’ later bedtimes as well as nighttime media consumption and electronic device use [1]. Due to their parents’ lifestyle and their own behavior, children have adopted a night-oriented lifestyle [20]. Moreover, the body’s sleep–wake cycle is susceptible to external factors [21]. It is important to judge appropriate sleep duration to ensure sleep quality. Focusing on sleep quality, measures are needed to encourage children to establish a sleep–wake rhythm [21]. Clarifying the social determinants of children’s sleep habits can inform policy discussions targeting children and their parents from a life-course epidemiology perspective [6]. Children’s and parents’ sleep habit-related factors have conventionally been analyzed separately [20,21]. We found no research exploring multiple factors within a single, broad-based investigation, nor any work holistically assessing children’s sleep habits in terms of amount, characteristics, and quality. This study aimed to explore four key aspects of elementary students’ sleep habits—bedtime, wake-up time, sleep duration, and daytime sleepiness—to identify associations among lifestyle habits and familial and social factors, and to determine the relative importance of these indicators.

## 2. Materials and Methods

### 2.1. Participants and Survey Outline

The Super Shokuiku School Project was designed to investigate food education and was supported by the Japanese Ministry of Education, Culture, Sports, Science and Technology. The overall purpose of the project was to promote healthy lifestyles in schoolchildren and to improve their health. For the purpose of evaluating the project, 3 questionnaire surveys were conducted among 5 participating elementary schools. The baseline survey (Phase 1) was conducted before food education in May 2014. The follow-up surveys were conducted after food education in December 2014 (Phase 2) and January 2016 (Phase 3). This study conducted a cross-sectional study using Phase 3 findings that closely investigated parents’ lifestyles, along with children’s lifestyles and sleep habits. A total of 2129 children aged between 6 and 13 years who attended 1 of 5 elementary schools in the city of Takaoka, Toyama Prefecture, Japan, participated in this study. Twenty of the children’s parents cannot read Japanese, thus they were excluded from the survey. A total of 1986 children agreed to participate in our survey and returned the questionnaires (response rate: 93.3%). The survey was approved by the institutional review board for the University of Toyama. The research objective and survey details were explained by the teachers, and informed consent was obtained from both children and parents prior to their participation.

### 2.2. Questionnaire

We used child lifestyle and social and family factors approved in the previous cohort and Super Shokuiku Project studies [15,17,22]. The categories for the organization of answers were determined in a previous study [6,15,17,22]. Participants were asked to complete questionnaires asking about sex, grade (age 6–13), sleep habits, lifestyle, social background, and parental lifestyle. Children responded to items concerning sex, grade (age 6–13), sleep habits, and their lifestyles, while their parents responded to items concerning the social background and parental lifestyle (Appendix A). All completed questionnaires were returned to the respondents’ respective schools.

#### 2.2.1. Sleep Habits

Child sleep habits on weekdays were assessed based on bedtime, wake-up time, nighttime sleep duration, and daytime sleepiness. Here, “bedtime” [6] was classified into two categories: “<22:00” and “≥22:00”. Further, “wake-up time” [6] was classified into the categories of “<07:00” and “≥07:00”, while “nighttime sleep duration” [17,22,23] was categorized according to the number of hours slept (i.e., “≥8 h” and “<8 h”). Finally, “daytime sleepiness” was classified into the categories of “no” and “yes.” Japanese elementary school children averaged approximately 8.5 h of sleep per night [24]. We examined the validity of the hours slept by young children as reported by their parents [23]. “Bedtime,” “wake-up time,” and “nighttime sleep duration” were significantly related to data recorded by Actiwatch devices [23,25]. Gaina et al. [26] reported that a sleep time interval of more than 8 h showed no increase in the risk of sleepiness. Sekine [6] reported that going to bed after 22:00 and waking up after 07:00 dramatically increased the risk of other negative lifestyle habits among 4th grade (aged 9–10 years) elementary school students. Based on this evidence, “bedtime ≥22:00,” “wake-up time ≥07:00,” “sleep duration <8 h,” and “daytime sleepiness” were defined as poor sleep habits to establish criteria to holistically assess the amount, characteristics, and quality of children’s sleep.

#### 2.2.2. Child Lifestyle Factors

Child lifestyles were assessed based on breakfast consumption, screen time h/day, and frequency of physical activity. Here, “breakfast consumption” [17,22,27] was classified into “eat every day” and “skipping,” which was defined as not eating breakfast every day [17,22]. Next, “screen time h/day” was measured for weekdays and included television and film viewing, gaming, and Internet use [17,18]. Responses were given according to a 6-point scale and divided into 3 categories [17]: “<2 h (none, <1 h, and <2 h),” “2–3 h (<3 h),” and “≥3 h (3 to <4 h and ≥4 h).” The Japan Pediatric Association recommends that total screen time be limited to <2 h/day [14], and Gaina et al. [26] reported that a TV viewing time of 3 h/day or more significantly affected children’s sleepiness. We defined a TV viewing time of 3 h/day or more as excessive. “Frequency of activity” [22] was answered according to a 4-point scale and divided into “Often (very often, often)” or “Not often (rarely, almost never).” We did not record responses for weekdays and holidays. The validity of both items was determined by a previous study [28]. A high frequency of physical activity was significantly associated with an increase in energy expenditure originating from physical activity [28].

#### 2.2.3. Social Background and Parental Lifestyle

Social background was assessed based on household, self-perceived family affluence, mother’s employment status, and parental health behaviors (i.e., mother and father) [29]. “Household” [12,22,23] was categorized as either “three-generation family” or “nuclear family (single parent included),” while “socioeconomic status” was determined according to perceived “family affluence” [12,15,18]. “Family affluence” was assessed according to 2 response categories [15], divided into “very affluent, affluent, neither” or “not much affluent, not affluent.” Further, “mother’s employment status” [12,15,18] included 3 response categories [15], divided into “full-time or part-time” and “unemployed (housewives).” Breslow’s 7 health practices are widely acknowledged and were, thus, used as parental health indicators [29]. Examined behaviors included (1) adequate sleep time, (2) not smoking, (3) appropriate weight control, (4) not drinking excessively, (5) regular physical activity, (6) not skipping breakfast, and (7) not snacking frequently. Parents answered “yes” or “no” to the 7 items; “yes” responses were summed to provide cumulative behavior scores ranging from 0 to 7. We divided respondents into 3 groups [15,17,22], as in previous research, based on their reported number of healthy behaviors: “poor (0–3),” “moderate (4–5),” and “good (6–7),” with higher scores indicating more healthy behaviors.

### 2.3. Statistical Analyses

Correlations between independent variables were examined using Spearman rank correlation coefficients. Logistic regression analyses were conducted to evaluate the strengths of the associations between poor child sleep habits and the items of social background, parental lifestyle, and child lifestyle. All variables were simultaneously entered into the model during multivariate analyses. Odds ratios (ORs) and 95% confidence intervals (CIs) were also calculated. We conducted the Hosmer−Lemeshow test to examine the goodness of fit. Since the quantity and quality of sleep were interrelated [30], we developed a model that does not consider the quantity and another that does. In logistic regression analysis, the sample size required for groups with a smaller outcome was determined by multiplying the number of factors by 10 [31]. Since the number of factors in this study was 10, the minimum required smaller outcome sample size was 100. Of the 4 outcomes of this study, the wake-up time (*n* = 85) was slightly below this requirement, but the sample size was sufficient for the other variables.

All analyses were conducted using the Statistical Package for the Social Sciences (SPSS) software version 26.0 J (IBM, Armonk, NY, USA). Two-tailed *p*-values less than 0.05 were considered statistically significant for all tests.

## 3. Results

Table 1 shows the participants’ characteristics. Of the 2129 respondents who returned their questionnaires, 1882 (88.4%; 943 boys and 939 girls) answered all relevant items, and their data were included for analysis. Child “bedtime ≥22:00” was reported by 28.9% of respondents (*n* = 544), 4.5% (*n* = 85) reported “wake-up time ≥07:00,” 21.5% (*n* = 404) reported “<8 h sleep duration,” and 17.0% (*n* = 320) reported “daytime sleepiness.” Regarding child lifestyle factors, the most frequent answers to the items assessing eating breakfast, screen time h/day, and frequency of physical activity, were “eating every day,” “<2 h/day,” and “often,” respectively. Regarding social background factors, the most frequent answers to the items assessing household, perceived family affluence, and mother’s employment status were “nuclear family,” “affluent,” and “employed,” respectively. Regarding parental lifestyle, the most frequent answers to the questions assessing parental health behaviors according to Breslow and Enstrom [29] were “father exhibits a poor number of health behaviors (0–3)” and “mother exhibits a moderate number of health behaviors (4–5).”

Table 2 shows the results of logistic regression analyses conducted to determine the strengths of the associations of “bedtime ≥22:00” with social background and child lifestyle. The multivariate analysis revealed the following results: “bedtime ≥22:00” correlated with girls (adjusted odds ratio [OR] 1.32; 95% confidence interval [CI] 1.05–1.66), the 2nd through 6th grades (ages 7–13), skipping breakfast (adjusted OR 2.97; 95% CI, 2.02–4.37), prolonged screen time (≥3 h/day; adjusted OR 1.70; 95% CI, 1.18–2.44), and poor Breslow’s health behaviors among mothers (adjusted OR 1.75; 95% CI, 1.23–2.49).

Table 3 shows the results of logistic regression analyses conducted to determine the strengths of the associations of “wake-up time ≥07:00” with social background and child lifestyle. The multivariate analysis revealed the following results: “wake-up time ≥07:00” correlated with children who skipped breakfast (adjusted OR 5.45; 95% CI, 3.20–9.30) and those who reported prolonged screen time (≥3 h/day; adjusted OR 3.05; 95% CI, 1.64–5.67).

Table 4 shows the results of logistic regression analyses conducted to determine the strengths of the associations of “<8 h sleep duration” with social background and child lifestyle. The multivariate analysis revealed the following results: “<8 h sleep duration” correlated with children in the 3rd through 6th grades (ages 8–13), skipping breakfast (adjusted OR 1.61; 95% CI, 1.09–2.36), and mothers being employed (adjusted OR 1.53; 95% CI, 1.04–2.25).

Table 5 shows the results of logistic regression analyses conducted to determine the strengths of the associations of “daytime sleepiness” with social background and child lifestyle. In Model 1, the multivariate analysis revealed the following results: “daytime sleepiness” correlated with children in the 5th through 6th grades (ages 10–13), skipping breakfast (adjusted OR 2.13; 95% CI, 1.46–3.12), lack of physical activity (adjusted OR 1.42; 95% CI, 1.09–1.85), moderate Breslow’s health behaviors among mothers (adjusted OR 1.45; 95% CI, 1.03–2.03), and poor Breslow’s health behaviors among mothers (adjusted OR 2.10; 95% CI, 1.41–3.12). In Model 2, the inclusion of sleep duration—known to be predictive of daytime sleepiness—as a moderating variable eliminated the significance of the heightened risk of daytime sleepiness in children in the 5th and 6th grades (ages 10–13). Nonetheless, the following associations remained significant, as in Model 1, demonstrating that they independently contribute to daytime sleepiness: Skipping breakfast (adjusted OR 2.01; 95% CI, 1.37–2.96), lack of physical activity (adjusted OR 1.43; 95% CI, 1.10–1.88), moderate Breslow’s health behaviors among mothers (adjusted OR 1.43; 95% CI, 1.01–2.01), and poor Breslow’s health behaviors among mothers (adjusted OR 2.05; 95% CI, 1.37–3.05).

In Table 1, Table 2, Table 3, Table 4 and Table 5, the correlation coefficients between pair-wise independent variables ranged from −0.113 to 0.325, indicating no multicollinearity. Results of the Hosmer-Lemeshow test validated the models.

## 4. Discussion

Sleep quantity and quality have decreased considerably among young people in recent decades [19,32]. This study aims at a holistic assessment of how the sleep habits of elementary school children—as indicated by bedtime, wake-up time, sleep duration, and daytime sleepiness—are associated with lifestyle, familial, and social factors. We found that children whose mothers practiced poor health behaviors were more likely to go to bed late and feel sleepy during the daytime compared to children whose mothers practiced good health behaviors. Our results also support previous claims that children’s sleep habits are closely linked to the mother’s lifestyle habits and employment [14,17,33], seemingly reflecting the mother’s dominant role in parenting. Children with employed mothers were more likely to sleep less than the recommended eight hours compared to the children of housewives. However, neither parent nor maternal employment had significant associations with wake-up time. Past research has shown that children of fully-employed mothers spend more time using electronic devices and in front of screens [12,17] and that they frequently snack between meals [14], implicating children’s evening and nighttime habits at home in their sleep loss. Educational interventions aimed at improving children’s sleep habits must also target improvement in parents’ lifestyle habits and incorporate strategies to work with and provide support to mothers in the community and at home.

Skipping breakfast was the only lifestyle habit assessed that was consistently and significantly associated with all four sleep indicators: Bedtime, wake-up time, sleep duration, and daytime sleepiness. Short sleep duration and prolonged screen time have been identified to contribute to this behavior in elementary school students [15]. Nighttime exposure to blue light suppresses the secretion of melatonin, causing difficulties falling asleep and getting up [34]. Research for Health Behavior in European Children showed that adolescents exceeding 2 h/day of screen time had 20% higher odds of reporting sleep-onset difficulties [32]. We found that excessive screen time, defined here as three or more hours per day, and skipping breakfast were independently linked to later bed and wake-up times. However, neither of the two lifestyle habits were associated with sleep duration. This pattern suggests that children could experience a nocturnal shift in their daily routines if they use electronic devices or consume media for more than 3 h/day, leading them to miss breakfast. A healthy breakfast is imperative for a child’s good physical health and performance at school [35]. With the enactment of the Basic Act on Food Education in Japan in 2005 [36], the government began promoting nutritional education using the slogan “Early to bed, early to rise, and breakfast” [37]. Our findings suggest that initiatives to improve children’s sleep habits would be most effective if they adopted a more holistic approach, expanding their scope beyond sleep habits to include lifestyle improvements, such as reducing nighttime screen time and ensuring breakfast consumption.

Children in the 5th and 6th grades (aged 10–13 years) were significantly more likely to experience daytime sleepiness compared with 1st graders (aged 6–7); however, no such significant associations with grade were observed for wake-up time. A higher OR of daytime sleepiness among 5th and 6th graders (aged 10–13) was attenuated when sleep duration was included in the model, but the OR of short sleep duration remained significant; this suggests that short sleep duration is a predictor of daytime sleepiness in these grades.

Past research has found that students who feel sleepy engage in little physical activity during the day [23]. Normal secretion of the neurochemicals serotonin and melatonin is crucial to obtaining high-quality sleep. Serotonin release is more prominent during the day, encouraging wakefulness and activity. However, at night, the balance shifts to melatonin, a crucial regulator of the circadian clock, promoting feelings of sleepiness. Physical inactivity is known to blunt the activity of serotonergic neurons [21], presenting a possible mechanism for its contribution to daytime sleepiness. Physical activity has been associated with better health and performance [38,39], thus ensuring that elementary school students regularly engage in moderate physical activity could greatly help to reduce the incidence of daytime sleepiness in this population. We believe this study’s findings can be useful to inform discussions of educational interventions designed to improve elementary school students’ sleep habits.

This study design was cross-sectional and correlational, limiting the ability to infer causation. Given the great diversity in lifestyle habits among parents and children, other factors not considered here may contribute to children’s night-oriented lifestyle. Moreover, this study’s findings may simply reflect local trends unique to the provincial city in which it was conducted. Future research must further investigate these matters by accounting for the influence of how children and their parents spend their time after school and at night, in addition to potentially region-specific characteristics.

## 5. Conclusions

This study was a holistic assessment of how the sleep habits of elementary school children—as indicated by sleep duration, bedtime, wake-up time, and daytime sleepiness—could be associated with lifestyle, familial, and social factors. Children’s sleep habits were associated with the lifestyle habits (specifically, the health behaviors) of both parents, as well as the employment status of the mother. In addition, poor sleep habits were frequently associated with three factors related to child lifestyle: Skipping breakfast, excessive screen time, and infrequent physical activity. Notably, media consumption and electronic device usage of three or more hours per day reflected an overall nocturnal shift in children’s routines, rather than an overall reduction in sleep, as evidenced by this habit’s associations with later bed and wake-up times. Efforts to improve children’s sleep habits should involve collaboration with communities and families to address the issues posed by parental lifestyle and employment status, and to regularize their life rhythms through a combination of food education, decreased screen time, and regular exercise. This study’s findings can usefully inform discussions of educational interventions designed to improve the sleep habits of elementary school students.

## Figures and Tables

**Table 1 children-08-00110-t001:** Participants’ characteristics.

Variable		*n*	%
Sex	Boys	943	50.1
	Girls	939	49.9
Grade	1st (age 6–7)	318	16.9
	2nd (age 7–8)	301	16.0
	3rd (age 8–9)	314	16.7
	4th (age 9–10)	280	14.9
	5th (age 10–11)	328	17.4
	6th (age 11–13)	341	18.1
Bedtime	<22:00	1338	71.1
	≥22:00	544	28.9
Wake-up time	<7:00	1797	95.5
	≥7:00	85	4.5
Sleep duration	≥8 h	1478	78.5
	<8 h	404	21.5
Daytime sleepiness	No	1562	83.0
	Yes	320	17.0
Eating breakfast	Every day	1731	92.0
	Skipping	151	8.0
Screen time, h/day	<2	1120	59.5
	2–3	562	29.9
	≥3	200	10.6
Frequency of physical activity	Often	1370	72.8
	Not often	512	27.2
Household	Three-generation family	608	32.3
	Nuclear family	1274	67.7
Perceived family affluence	Affluence	1366	72.6
	Not affluent	516	27.4
Mother’s employment status	Unemployed (housewives) Employed	259	13.8
		1623	86.2
Father’s Breslow’s seven health practice score	Good (6–7)	368	19.6
	Moderate (4–5)	731	38.8
	Poor (0–3)	783	41.6
Mother’s Breslow’s seven health practice score	Good (6–7)	485	25.8
	Moderate (4–5)	1015	53.9
	Poor (0–3)	382	20.3

**Table 2 children-08-00110-t002:** Associations of social, family, and lifestyle factors with late bedtimes of children.

	≥22:00 *n*(%)	Univariate OR(95% CI)	Multivariate OR(95% CI)
Sex			
Boys	253(26.8)	1	1
Girls	291(31.0)	1.22(1.00–1.50) *	1.32(1.05–1.66) *
Grade			
1st (age 6–7)	17(5.3)	1	1
2nd (age 7–8)	36(12.0)	2.41(1.32–4.38) †	2.41(1.31–4.43) †
3rd (age 8–9)	57(18.2)	3.93(2.23–6.92) ‡	3.81(2.14–6.77) ‡
4th (age 9–10)	93(33.2)	8.81(5.09–15.2) ‡	8.46(4.85–14.8) ‡
5th (age 10–11)	144(43.9)	13.9(8.12–23.7) ‡	13.4(7.78–23.1) ‡
6th (age 11–13)	197(57.8)	24.2(14.2–41.3) ‡	24.1(14.0–41.4) ‡
Eating breakfast			
Every day	462(26.7)	1	1
Skipping	82(54.3)	3.26(2.33–4.57) ‡	2.97(2.02–4.37) ‡
Screen time h/day			
<2	278(24.8)	1	1
2–3	175(31.1)	1.37(1.09–1.71) †	1.16(0.90–1.50)
≥3	91(45.5)	2.53(1.86–3.45) ‡	1.70(1.18–2.44) †
Frequency of physical activity			
Often	378(27.6)	1	1
Not often	166(32.4)	1.26(1.01–1.57) *	0.98(0.77–1.26)
Household			
Three-generation family	184(30.3)	1	1
Nuclear family	360(28.3)	0.91(0.73–1.12)	0.93(0.73–1.18)
Perceived family affluence			
Affluence	377(27.6)	1	1
Not affluent	167(32.4)	1.26(1.01–1.56) *	1.14(0.89–1.46)
Mother’s employment status			
Housewives	56(21.6)	1	1
Employed	488(30.1)	1.56(1.14–2.13) †	1.27(0.89–1.82)
Father’s Breslow’s seven health practice score			
Good (6–7)	93(25.3)	1	1
Moderate (4–5)	208(28.5)	1.18(0.88–1.56)	1.02(0.74–1.41)
Poor (0–3)	243(31.0)	1.33(1.01–1.76) *	0.94(0.67–1.31)
Mother’s Breslow’s seven health practice score			
Good (6–7)	114(23.5)	1	1
Moderate (4–5)	288(28.4)	1.29(1.00–1.66) *	1.27(0.96–1.69)
Poor (0–3)	142(37.2)	1.93(1.43–2.59) ‡	1.75(1.23–2.49) †

OR, odds ratio; 95% CI, 95% confidence interval, * *p* < 0.05, † *p* < 0.01, ‡ *p*< 0.001.

**Table 3 children-08-00110-t003:** Associations of social, family, and lifestyle factors with late wake-up time of children.

	≥7:00 *n*(%)	Univariate OR(95% CI)	Multivariate OR(95% CI)
Sex			
Boys	50(5.3)	1	1
Girls	35(3.7)	0.69(0.44–1.08)	0.74(0.46–1.19)
Grade			
1st (age 6–7)	16(5.0)	1	1
2nd (age 7–8)	8(2.7)	0.52(0.22–1.22)	0.43(0.18–1.05)
3rd (age 8–9)	12(3.8)	0.75(0.35–1.61)	0.57(0.26–1.26)
4th (age 9–10)	11(3.9)	0.77(0.35–1.69)	0.55(0.24–1.24)
5th (age 10–11)	13(4.0)	0.78(0.37–1.65)	0.52(0.24–1.14)
6th (age 11–13)	25(7.3)	0.79(0.78–2.85)	1.07(0.54–2.13)
Eating breakfast			
Every day	57(3.3)	1	1
Skipping	28(18.5)	6.69(4.10–10.9) ‡	5.45(3.20–9.30) ‡
Screen time h/day			
<2	32(2.9)	1	1
2–3	28(5.0)	1.78(1.06–2.99) *	1.50(0.87–2.58)
≥3	25(12.5)	4.86(2.81–8.39) ‡	3.05(1.64–5.67) ‡
Frequency of physical activity			
Often	61(4.5)	1	1
Not often	24(4.7)	1.06(0.65–1.71)	0.96(0.57–1.61)
Household			
Three-generation family	21(3.5)	1	1
Nuclear family	64(5.0)	1.48(0.89–2.44)	1.34(0.79–2.27)
Perceived family affluence			
Affluence	52(3.8)	1	1
Not affluent	33(6.4)	1.73(1.10–2.70) *	1.48(0.91–2.39)
Mother’s employment status			
Housewives	6(2.3)	1	1
Employed	79(4.9)	2.16(0.93–5.00)	1.91(0.80–4.54)
Father’s Breslow’s seven health practice score			
Good (6–7)	13(3.5)	1	1
Moderate (4–5)	28(3.8)	1.09(0.56–2.13)	1.11(0.55–2.25)
Poor (0–3)	44(5.6)	1.63(0.86–3.06)	1.08(0.54–2.18)
Mother’s Breslow’s seven health practice score			
Good (6–7)	19(3.9)	1	1
Moderate (4–5)	41(4.0)	1.03(0.59–1.80)	0.85(0.47–1.53)
Poor (0–3)	25(6.5)	1.72(0.93–3.17)	0.94(0.47–1.89)

OR, odds ratio; 95% CI, 95% confidence interval, * *p* < 0.05, ‡ *p* < 0.001.

**Table 4 children-08-00110-t004:** Associations of social, family, and lifestyle factors with sleep duration of children.

	<8 h *n*(%)	Univariate OR(95% CI)	Multivariate OR(95% CI)
Sex			
Boys	192(20.4)	1	1
Girls	212(22.6)	1.14(0.92–1.42)	1.14(0.90–1.43)
Grade			
1st (age 6–7)	27(8.5)	1	1
2nd (age 7–8)	38(13.6)	1.56(0.93–2.62)	1.55(0.92–2.61)
3rd (age 8–9)	45(14.3)	1.80(1.09–2.99) *	1.77(1.06–2.94) ‡
4th (age 9–10)	61(21.8)	3.00(1.85–4.88) ‡	2.82(1.73–4.60) ‡
5th (age 10–11)	102(31.1)	4.86(3.08–7.69) ‡	4.62(2.91–7.35) ‡
6th (age 11–13)	131(38.4)	6.72(4.28–10.6) ‡	6.53(4.14–10.3) ‡
Eating breakfast			
Every day	355(20.5)	1	1
Skipping	49(32.5)	1.86(1.30–2.67) ‡	1.61(1.09–2.36) *
Screen time h/day			
<2	225(20.1)	1	1
2–3	126(22.4)	1.15(0.90–1.47)	0.99(0.76–1.29)
≥3	53(26.5)	1.43(1.01–2.03) *	1.00(0.69–1.47)
Frequency of physical activity			
Often	286(20.9)	1	1
Not often	118(23.0)	1.14(0.89–1.45)	0.99(0.76–1.28)
Household			
Three-generation family	138(22.7)	1	1
Nuclear family	266(20.9)	0.90(0.71–1.13)	0.93(0.73–1.19)
Perceived family affluence			
Affluent	281(20.6)	1	1
Not affluent	123(23.8)	1.21(0.95–1.54)	1.12(0.87–1.45)
Mother’s employment status			
Housewives	37(14.3)	1	1
Employed	367(22.6)	1.75(1.21–2.53)	1.53(1.04–2.25) *
Father’s Breslow’s seven health practice score			
Good (6–7)	63(17.1)	1	1
Moderate (4–5)	162(22.2)	1.38(1.00–1.90)	1.25(0.89–1.75)
Poor (0–3)	179(22.9)	1.43(1.04–1.97) *	1.17(0.82–1.66)
Mother’s Breslow’s seven health practice score			
Good (6–7)	86(17.7)	1	1
Moderate (4–5)	220(21.7)	1.28(0.97–1.69)	1.22(0.91–1.64)
Poor (0–3)	98(25.7)	1.60(1.15–2.22) *	1.43(0.99–2.06)

OR, odds ratio; 95% CI, 95% confidence interval, * *p* < 0.05, ‡ *p* < 0.001.

**Table 5 children-08-00110-t005:** Associations of social, family, and lifestyle factors with daytime sleepiness of children.

	Yes *n*(%)	Univariate OR(95% CI)	Multivariate OR(95% CI)	Multivariate OR(95% CI)
			**Model 1**	**Model 2**
Sex				
Boys	153(16.2)	1	1	1
Girls	167(17.8)	1.12(0.88–1.42)	1.13(0.88–1.45)	1.12(0.87–1.45)
Grade				
1st (age 6–7)	40(12.6)	1	1	1
2nd (age 7–8)	36(12.0)	0.94(0.58–1.53)	0.96(0.59–1.57)	0.93(0.57–1.52)
3rd (age 8–9)	45(14.3)	1.16(0.74–1.84)	1.12(0.70–1.78)	1.05(0.66–1.69)
4th (age 9–10)	51(18.2)	1.55(0.99–2.43)	1.42(0.90–2.26)	1.29(0.81–2.05)
5th (age 10–11)	72(22.0)	1.95(1.28–2.98) †	1.81(1.17–2.79) †	1.52(0.98–2.37)
6th (age 11–13)	76(22.3)	1.99(1.31–3.03) ‡	1.82(1.19–2.80) †	1.43(0.92–2.23)
Eating breakfast				
Every day	270(15.6)	1	1	1
Skipping	50(33.1)	2.68(1.86–3.85) ‡	2.13(1.46–3.12) ‡	2.01(1.37–2.96) ‡
Screen time h/day				
<2	163(14.6)	1	1	1
2–3	108(19.2)	1.40(1.07–1.83) *	1.17(0.88–1.55)	1.18(0.89–1.56)
≥3	49(24.5)	1.91(1.33–2.74) ‡	1.27(0.86–1.89)	1.28(0.86–1.90)
Frequency of physical activity				
Often	208(15.2)	1	1	1
Not often	112(21.9)	1.56(1.21–2.02) ‡	1.42(1.09–1.85) †	1.43(1.10–1.88) †
Household				
Three-generation family	100(16.4)	1	1	1
Nuclear family	220(17.3)	1.06(0.82–1.37)	1.05(0.80–1.37)	1.06(0.81–1.39)
Perceived family affluence				
Affluent	212(15.5)	1	1	1
Not affluent	108(20.9)	1.44(1.11–1.86) †	1.27(0.97–1.66)	1.26(0.96–1.65)
Mother’s employment status				
Housewives	38(14.7)	1	1	1
Employed	282(17.4)	1.22(0.85–1.77)	1.09(0.74–1.59)	1.05(0.71–1.54)
Father’s Breslow’s seven health practice score				
Good (6–7)	50(13.6)	1	1	1
Moderate (4–5)	106(14.5)	1.08(0.75–1.55)	0.94(0.64–1.36)	0.90(0.62–1.32)
Poor (0–3)	164(20.9)	1.69(1.19–2.38) †	1.14(0.78–1.66)	1.11(0.76–1.62)
Mother’s Breslow’s seven health practice score				
Good (6–7)	54(11.1)	1	1	1
Moderate (4–5)	169(16.7)	1.59(1.15–2.21) †	1.45(1.03–2.03) *	1.43(1.01–2.01) *
Poor (0–3)	97(25.4)	2.72(1.89–3.91) ‡	2.10(1.41–3.12) ‡	2.05(1.37–3.05) ‡
Sleep duration				
≥8 h	204(13.8)	1		1
<8 h	116(28.7)	2.52(1.94–3.27) ‡		2.11(1.59–2.78) ‡

OR, odds ratio; 95% CI, 95% confidence interval, * *p* < 0.05, † *p* < 0.01, ‡ *p* < 0.001. Model 1: Does not include sleep duration, Model 2: Includes sleep duration.

## Data Availability

The data we used to derive our findings are unsuitable for public deposition due to ethical restrictions and specific legal framework in Japan. It is prohibited by the Act on the Protection of Personal Information (Act No.57 of 30 May 2003, amended on 9 September 2015) to publicly deposit data containing personal information. The Ethical Guidelines for Epidemiological Research enforced by the Japan Ministry of Education, Culture, Sports, Science, and Technology and the Ministry of health, Labor and Welfare also restrict the open sharing of the epidemiologic data.

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
