# Peer review of "Social and Family Factors as Determinants of Sleep Habits in Japanese Elementary School Children: A Cross-Sectional Study from the Super Shokuiku School Project"

_children, 2021, doi:10.3390/children8020110_

Round 1
Reviewer 1 Report
In revised version of this paper used in that research questionnaire should be provided.
It could be putted into body of the paper (with translation into English) as figure.
Also the minimum number of participants of this kind of research should be calculated and provided. About 2k participant of the research seams to be huge number, but this number must be compared with minimum number of people needed for this kind of research.
Also at some places of the paper the age of participants together with their grade should be provided. At some places this information is given but at some not andl only information about grades is provided.
Please describe children age not only be grades but also but their age.
Author Response
Response to Reviewer 1:
We wish to express our appreciation to the Reviewer for their insightful comments, which helped to significantly improve the paper. Changes to the manuscript in response to Reviewer 1 are shown in orange font. Changes in response to other reviewers are shown in pink and blue. Finally, the manuscript has been edited by a native speaker of English, who is also a professional academic editor. Corrections in the revised manuscript are shown in green font.
- In revised version of this paper used in that research questionnaire should be provided. It could be putted into body of the paper (with translation into English) as figure.
Response: We added the research questionnaire to Supplementary Materials. Please see the attachment.
We added the following sentence to the manuscript (p. 12, lines 349-350):
Supplementary Materials: Research questionnaire in the Super Shokuiku School Project (Phase 3)
- Also the minimum number of participants of this kind of research should be calculated and provided. About 2k participant of the research seams to be huge number, but this number must be compared with minimum number of people needed for this kind of research.
Response: We added the following sentence to the manuscript (p. 4, lines 167-173):
In logistic regression analysis, the sample size required for groups with a smaller outcome is determined by multiplying the number of factors by 10 [31]. Since the number of factors in this study was 10, the minimum required smaller outcome sample size was 100. Of the four outcomes of this study, the wake-up time (n=85) was slightly below this requirement, but the sample size was sufficient for the other variables.
We added the following reference to the manuscript. (p. 15, lines 446-448):
- Peduzzi, P.; Concato, J.; Kemper, E.; Holford, T. R.; Feinstein, A. R. A simulation study of the number of events per variable in logistic regression analysis. J Clin Epidemiol. 1996, 49 (12), 1373–1379; doi: https://doi.org/10.1016/S0895-4356(96)00236-3.
- Also at some places of the paper the age of participants together with their grade should be provided. At some places this information is given but at some not and only information about grades is provided. Please describe children age not only be grades but also but their age.
Response: We revised the manuscript to clarify the ages differentiate between age groups of the participating children, as demonstrated in the example below. We also ensured that the age differences were clear in the tables.
Example:
Children in the 5th and 6th grades (aged 10-13 years) were significantly more likely to experience daytime sleepiness compared with 1st graders (aged 6-7); however, no such significant associations with grade were observed for wake-up time. (p. 11, lines 302-306)

Reviewer 2 Report
Good study.
The study explored the associations of lifestyle, familial, and social factors with key aspects of sleep habits in children.
The sample is excelent.
The materials and the results are clears.
Limitation: the study is correlational
Author Response
Response to Reviewer 2:
We wish to express our appreciation to the Reviewer for their insightful comments, which helped to significantly improve the paper. Changes to the manuscript in response to Reviewer 2 are shown in pink font. Changes in response to other reviewers are shown in orange and blue. Finally, the manuscript has been edited by a native speaker of English, who is also a professional academic editor. Corrections in the revised manuscript are shown in green font.
- Good study.
The study explored the associations of lifestyle, familial, and social factors with key aspects of sleep habits in children.
The sample is excellent.
The materials and the results are clears.
Limitation: the study is correlational
Response:
Thank for your positive feedback. In response to your comment, we added the following limitation (p. 12, lines 322):
This study design was cross-sectional and correlational, limiting ability to infer causation.

Reviewer 3 Report
See uploaded document.

Author Response
Response to Reviewer 3:
We wish to express our appreciation to the Reviewer for their insightful comments, which helped to significantly improve the paper. Changes to the manuscript in response to Reviewer 3 are shown in blue font. Changes in response to other reviewers are shown in orange and pink. Finally, the manuscript has been edited by a native speaker of English, who is also a professional academic editor. Corrections in the revised manuscript are shown in green font.
- Overall, the manuscript is well-written and clear. The scope of the study is appropriate, and the conclusions are couched in the literature, with clear distinction between association and causation. There are three points on which I would like to see additional clarity and explanation.
First, the authors do not provide the rationale for collapsing responses into categories (e.g., six screen time responses recorded as three categories, three mother’s employment responses recorded as two categories). Why was this done for the above-referenced variables and others?
Response: Thank for your constructive feedback. We used child lifestyles and social and family factors approved in previous cohort and Super Shokuiku Project studies [15,17,22]. The categories for organization of answers were determined in a previous study.
First, we added the following sentence to the manuscript (p. 2, lines 93-94):
The categories for organization of answers were determined in a previous study [6,15,17,22].
Second, we added the following reference to the manuscript. (p. 3, lines 103, 106, 124, 126, 127, 144-147, 155):
- Second, as someone who does not use logistic regression that often, I did not completely understand (a) why both univariate and multivariate procedures were used, (b) what was gained from using both, and (c) what to take away from a given category having a significant result in univariate but not multivariate or vice versa.
First, in univariate analysis, we tested the significant difference between the dependent variable (poor sleep habits) and each independent variable. It was important to clarify the factors that are independently associated with children’s poor sleep habits, even after adjusting for potential confounding factors, including the social and family environment and children’s lifestyle factors. We used the results of multivariate analysis for consideration.
- Third, when analyzing the data for daytime sleepiness, the use of the two models is introduced rather late and not explained well. I would rather see this approach explained in the Methods with the other analysis descriptions. Further, I would like to know why this was the only case in which possible moderation / interaction between dependent variables was considered and why it was only considered in the multivariate analysis and not the univariate analysis.
Response: In response to your comment, we added the following sentence to the manuscript (p. 4, lines 166-167):
Since the quantity and quality of sleep are interrelated [30], we developed a model that does not consider quantity and another that does.
We added the following reference to the manuscript. (p. 15, lines 443-445):
- Sekine, M.; Tatsuse, T.; Cable, N.; Chandola, T.; Marmot, M. U-shaped associations between time in bed and the physical and mental functioning of Japanese civil servants: the roles of work, family, behavioral and sleep quality characteristics. Sleep Med. 2014, 15 (9), 1122–1131; doi: https://doi.org/10.1016/j.sleep.2014.04.012.
